# How Do Geriatric Scores Predict 1-Year Mortality in Elderly Patients with Suspected Pneumonia?

**DOI:** 10.3390/geriatrics6040112

**Published:** 2021-11-23

**Authors:** Alberto Nascè, Astrid Malézieux-Picard, Landry Hakiza, Thomas Fassier, Dina Zekry, Jérôme Stirnemann, Nicolas Garin, Virginie Prendki, Xavier Roux

**Affiliations:** 1Division of Internal Medicine for the Aged, Department of Rehabilitation and Geriatrics, Geneva University Hospitals, Chemin du Pont-Bochet 3, 1226 Thônex, Switzerland; astrid-marie.malezieux@hcuge.ch (A.M.-P.); landryrali.hakiza@hcuge.ch (L.H.); thomas.fassier@hcuge.ch (T.F.); dina.zekry@hcuge.ch (D.Z.); virginie.prendki@hcuge.ch (V.P.); xavier.roux@hcuge.ch (X.R.); 2Division of General Internal Medicine, Department of Internal Medicine, Rehabilitation and Geriatrics, Geneva University Hospitals, Rue Gabrielle-Perret-Gentil 4, 1205 Genève, Switzerland; jerome.stirnemann@hcuge.ch (J.S.); Nicolas.Garin@hopitalrivierachablais.ch (N.G.); 3Department of General Internal Medicine, Riviera Chablais Hospitals, Route du Vieux-Séquoïa 20, 1847 Rennaz, Switzerland; 4Division of Infectious Diseases, University Hospitals of Geneva, 1205 Genève, Switzerland; 5Division of Intensive Care Unit, University Hospitals of Geneva, 1205 Genève, Switzerland

**Keywords:** pneumonia, elderly, comorbidities, malnutrition, functionality, mortality

## Abstract

Background: Pneumonia has an impact on long-term mortality in elderly patients. The risk factors associated with poor long-term outcomes are understated. We aimed to assess the ability of scores that evaluate patients’ comorbidities (cumulative illness rating scale—geriatric, CIRS-G), malnutrition (mini nutritional assessment, MNA) and functionality (functional independence measure, FIM) to predict 1-year mortality in a cohort of older patients having a suspicion of pneumonia. Methods: Our prospective study included consecutive patients over 65 years old and hospitalized with a suspicion of pneumonia enrolled in a monocentric cohort from May 2015 to April 2016. Each score was analysed in univariate and multivariate models and logistic regressions were used to identify contributors to 1-year mortality. Results: 200 patients were included (51% male, mean age 83.8 ± 7.7). Their 1-year mortality rate was 30%. FIM (*p* < 0.01), CIRS-G (*p* < 0.001) and MNA (*p* < 0.001) were strongly associated with poorer long-term outcomes in univariate analysis. CIRS-G (*p* < 0.05) and MNA (*p* < 0.05) were significant predictors of 1-year mortality in multivariate analysis. Conclusion: Long-term prognosis of patients hospitalized for pneumonia was poor and we identified that scores assessing comorbidities and malnutrition seem to be important predictors of 1-year mortality. This should be taken into account for evaluating elderly patients’ prognosis, levels and goals of care.

## 1. Introduction

Pneumonia represents one of the greatest causes of hospitalization and mortality from infection in patients aged 65 or older and a major challenge for physicians [1]. Multiple morbidities and age-related conditions can interfere with the outcomes of pneumonia. Effective prognostication at patients’ admission is therefore required. The two most widely used and validated tools for determining pneumonia-related mortality are the pneumonia severity index (PSI) [2] and the confusion, urea, respiratory rate, blood pressure, and 65-years old score (CURB-65) [3]. It has also been shown that the sequential organ failure assessment score and its quick version (SOFA and qSOFA) [4,5] are good predictors of pneumonia prognosis [6]. However, these scores are only validated for short-term mortality and may have limitations in elderly patients because they do not estimate the biological reserve and systemic functionality, which contribute to their short- and especially long-term prognosis [7]. In this population, a more comprehensive approach, including a focus on patients’ comorbidities and their functional and nutritional status, may be preferrable in predicting their long-term prognosis.

Our research aimed to assess the ability of geriatric scores that evaluate patients’ comorbidities (cumulative illness rating scale—geriatric, CIRS-G) [8] malnutrition (mini nutritional assessment, MNA) [9] and functionality (functional independence measure, FIM) [10] to predict 1-year mortality in a cohort of older patients having a suspicion of pneumonia. These thorough and detailed scores were selected based on their frequent use in our clinical practice and performance reported previously in the literature [11]. They have also already demonstrated effectiveness in prognostic studies including geriatric patients suffering from several diseases, comprising pneumonia [12,13,14]. We used a prospective observational cohort including patients over 65 years with a suspicion of pneumonia.

## 2. Methods

### 2.1. Study Design and Participants

This study took place at a Department of Internal Medicine, Rehabilitation and Geriatrics in Switzerland, an 1800-bed tertiary care health institution serving a population of about 500,000 inhabitants.

Two hundred consecutive hospitalized patients aged 65 years or older suspected of pneumonia were enrolled in a prospective cohort study between the 1 May 2015 and the 30 April 2016, which aimed to determine whether low-dose computed tomography (LDCT) had the capacity to enhance the probability of diagnosing pneumonia in elderly patients, and is described elsewhere [15].

The clinical suspicion of community or hospital-acquired pneumonia (defined as an infection developing in 2 or more days after hospital admission) was based, in accordance with the Infectious Disease Society of America/American Thoracic Society (IDSA/ATS) guidelines, on the presence of at least 1 respiratory symptom (new or worsening cough, purulent sputum, pleuritic chest pain, new or worsening dyspnoea, respiratory rate >20 breaths/min, auscultatory findings or oxygen saturation <90% on room air) and at least 1 clinical or serological finding compatible with pneumonia (body temperature >38 °C or <35 °C, CRP > 10 mg/L, white blood cells (WBC) > 10 G/L with >85% neutrophils or band forms).

Patients who had been treated for a pulmonary infection during the previous 6 months, who had already undergone a CT scan during that specific episode or needed a contrast-enhanced CT, who had to be hospitalized in an intensive care unit (ICU), who had received antibiotics for more than 48 h before inclusion, or patients considered as unable to give their consent were excluded.

### 2.2. Data Collection

Data were retrieved from the electronic patient record system. For each included patient, collected data were categorized according to the following domains: demographics, previous treatments, laboratory findings, and comorbidities. The following scores were obtained prospectively within 48 h after admission: CURB-65, PSI, SOFA, qSOFA, MNA, and FIM. CIRS-G was retrieved retrospectively by two medical doctors and a research nurse. Dates of death were obtained by consulting the institutional database and the cantonal register of deaths. All scores are briefly described in the Appendix A.

### 2.3. Outcomes

In this research, we aimed to study the performance of comorbidities (CIRS-G), malnutrition (MNA), and functionality (FIM) assessment tools which were evaluated individually and in a multivariate analysis to predict 1-year mortality in elderly patients hospitalized for suspicion of pneumonia.

### 2.4. Statistical Analysis

Numbers and percentages are reported for categorical variables; medians and interquartile ranges (IQRs) are reported for continuous variables with non-normal distributions, whereas means and standard deviations (SDs) are reported for those continuous variables with a normal distribution.

Categorical variables were compared using the χ^2^ Test or Fisher’s exact test, while continuous variables were compared using the t-test or nonparametric Mann–Whitney U test.

Logistic regression analyses were used to examine associations between long-term mortality and risk factors. In the first step, each risk factor was tested individually. In the second step, risk factors showing an association in the univariate model (*p* < 0.1) were added to multivariate and adjusted models. To compare the accuracy of comorbidity, malnutrition, and functionality assessment tools (CIRS-G, MNA, and FIM) to predict mortality, and to standardize results according to pneumonia severity, the CURB-65 severity score was added into the logistic regression model. This five-point score, developed to stratify patients into different treatment groups depending on mortality risk, is one of the pneumonia severity scores proving some correlation with long-term prognosis in elderly patients and usually used in our clinical practice [16].

Prognostic values of the three studied scores demonstrating an individual capacity to predict 1-year mortality were compared using criteria of sensitivity, specificity, positive and negative likelihood ratios (LR), and diagnostic odds ratios (DOR). The most performing cut-offs were determined using the Youden index. Receiver operating characteristic (ROC) curves were equally performed and areas under the curves (AUC) measured. The Delong test was performed to compare AUC.

Analyses were performed using StataCorp. 2017 (Stata Statistical Software: Release 15. StataCorp LLC, College Station, TX, USA).

The study was carried out in accordance with the Declaration of Helsinki II Principles (W.M.Association, 2001) and was approved by the local ethics committee (CCER 14-250).

## 3. Results

A total of 200 patients were included in our study. Their mean age was 83.8 ± 7.8 years old; and 51% were male (Table 1). The overall 1-year mortality rate was 30%.

Univariate analysis found that the significant factors contributing to 1-year mortality were age, body mass index (BMI), urea, NT-proBNP and CURB-65, SOFA, FIM, CIRS-G, and MNA scores. Patients who died at one year were significantly older (*p* < *0*.005) and had a lower BMI (*p* < 0.001). Deceased patients had a higher CURB-65 score (*p* < 0.005, Table A1 shown in the Appendix A), a higher SOFA score (2.9 ± 2.1 vs. 2.3 ± 1.4, *p* < 0.05), a lower FIM score (76.3 ± 32 vs. 64.5 ± 26; *p* < 0.01), a higher CIRS-G score (25.5 ± 5.2 vs. 22.2 ± 6.5, *p* < 0.001) and a lower MNA score (6.8 ± 3.0 vs. 8.9 ± 2.6, *p* < 0.001) compared with 1-year survivors. No significant correlation was found for PSI or qSOFA scores. With regards to laboratory findings, patients who died at one year also had higher urea (10.4 ± 6.4 mmol/L vs. 8.9 ± 4.8 mmol/L; *p* < 0.05) and NT-proBNP levels (3402 ± 3032 ng/L vs. 2537 ± 2651 ng/L; *p* < 0.05). In multivariate analysis, only CIRS-G and MNA were found to be significant contributors to 1-year mortality (*p* < 0.05) (Table 2, Table A2 and Table A3, shown in the Appendix A).

The performance of all the scores proving to independently predict 1-year mortality were tested. Sensitivity and specificity were 59% and 70% for CIRS-G ≥ 26, 71% and 65% for MNA ≥ 8, and 58% and 52% for FIM ≥ 63, respectively (Table 3). The areas under the curves (AUC) values were 0.70 (95% CI 0.63–0.77) for MNA, 0.66 (95% CI 0.58–0.72) for CIRS-G, and 0.60 (95% CI 0.51–0.69) for FIM (Figure 1). No significant differences were found when we compared the AUC (results not shown).

## 4. Discussion

Through this study that evaluates the prognostic value of scores evaluating comorbidities, malnutrition and functionality in long-term mortality in an elderly population hospitalized with a suspicion of pneumonia, we found that CIRS-G and MNA were strong predictors of 1-year mortality.

Many studies reported a series of risk factors associated with long-term mortality in patients suffering from pneumonia [7]. Pulmonary infections may have significant impacts on various organ systems, such as respiratory, cardiovascular, and neurological ones, leading to the potential worsening of pre-existing comorbidities and subsequent higher fatality rates [7]. Therefore, a better understanding of long-term mortality prediction, measured at 30% in our study, seems urgent.

Amongst the risk factors commonly associated with poor long-term outcomes, we investigated the role of comorbidities, malnutrition and functionality.

An important aspect often playing a role in elderly patients’ mortality is malnutrition. The MNA was developed as a nutritional screening tool. Using this tool, we were able to identify a very strong correlation between malnutrition and poor outcomes at one year, indicating that assessment of the nutritional status at admission may help in reducing elderly patients’ mortality. Few other studies detected similar results. Yoon et al. [17], studying an elderly population with aspiration pneumonia, identified lower BMI and hypoalbuminemia as independent prognostic factors for 5-year mortality. Yeo et al. [18] recently highlighted that malnutrition was strongly linked with higher 2-year mortality in people suffering from pneumonia, particularly in the elderly, making essential a routine nutritional assessment at admission. Among elderly patients who have recovered from pneumonia, those who are malnourished have an increased risk of developing impaired muscle and respiratory function, which may lead to more severe long-term outcomes [19].

Regarding comorbidities, we took into consideration the cumulative illness rating score—geriatric (CIRS-G). One of our main findings was a strong correlation between the CIRS-G and mortality at one year. Similar mortality results, although not focused on a specific disease, were found in the recent literature. A systematic review on the performance of different morbidity scores to predict mortality in inpatients hospitalized for any medical condition showed that CIRS-G, as per 1 point increase, was significantly associated with post-discharge mortality [20]. Zekry et al. highlighted in patients hospitalized in an acute geriatric hospital that CIRS-G provided the most accurate risk prediction for 5-year mortality among six widely used multimorbidity scores [21]. Salvi et al. confirmed the validity of the CIRS-G as an indicator of health status and demonstrated its ability to predict 18-month mortality and rehospitalisation amongst elderly inpatients [22]. In the same line, Ritt et al. showed that CIRS-G proved accurate in forecasting 1-year mortality in elderly patients [23].

Amongst several scales estimating patients’ dependence and functionality [24,25,26], we used the FIM, which in a recent study among critically ill elderly patients admitted to an intermediate care unit proved a correlation between low ratings on the scale and higher 1-year mortality rates [27]. Another research showed that frailty, defined as unintentional weight loss, self-reported exhaustion, weakness, slow walking speed, and low physical activity, was strongly associated with the severity of pneumonia and higher 1-year mortality in older patients, suggesting that frailty should be detected early to improve their management [28]. In our findings, lower FIM ratings were individually associated with a poorer long-term prognosis but failed in finding significant 1-year mortality correlations in multivariate models. This could be explained by the role of confounders in the multivariate model, such as dementia, which is correlated with poor patients’ functionality.

In our study we also evaluated the capacity of the usual pneumonia and sepsis severity scores to predict 1-year mortality in our cohort of elderly patients with a suspicion of pneumonia. Interestingly, although these scores have been validated only for short-term outcomes, we found a correlation between CURB-65 and SOFA and 1-year mortality. It is noticeable that after adjustment with pneumonia and sepsis severity scores, the prognostic value of CIRS-G and MNA remain significant (Table 1 and Table A4 in the Appendix A).

The present study’s main strength was the consecutive inclusion of elderly patients hospitalized and treated for a suspicion of pneumonia. It has several limitations, however. As a single-centre study carried out with a relatively small number of patients, it should not be generalized to other hospitals. Since we focused our attention on elderly individuals with suspected pneumonia, our results should not be exported to other clinical contexts. Additionally, CIRS-G was retrieved retrospectively from medical records, exposing our analysis to potentials biases. Finally, we did not record the ‘do not resuscitate’ orders in our cohort, which might affect patients’ outcomes and be a source of bias.

## 5. Conclusions and Implications

The cumulative illness rating scale—geriatric (CIRS-G) and mini-nutritional assessment (MNA) were found to be promising prognostic tools for assessing long-term mortality in a cohort of elderly people suffering from suspected pneumonia. Our findings suggest that a more holistic approach, including nutritional and comorbidity assessments, should be performed when treating older patients with a suspicion of pneumonia and discussing their prognosis and goals of care. Nevertheless, further studies are needed to evaluate the concrete impact of managing comorbidities and malnutrition on long-term prognosis.

## Figures and Tables

**Figure 1 geriatrics-06-00112-f001:**
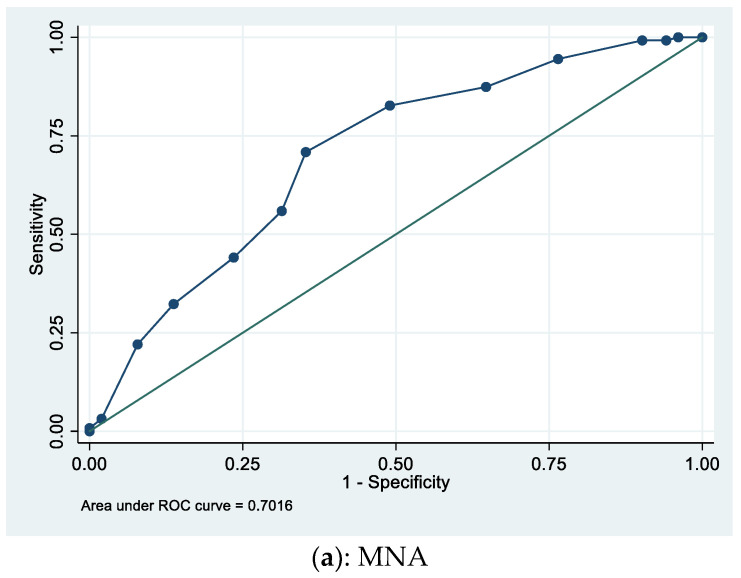
ROC (receiver operating characteristic) curves for pneumonia 1-year mortality: (**a**) MNA area under the curve = 0.70. Optimal cut-off point: 8; (**b**) CIRS-G area under the curve = 0.66. Optimal cut-off point: 26; (**c**) FIM area under the curve = 0.60. Optimal cut-off point: 64.

**Table 1 geriatrics-06-00112-t001:** Baseline characteristics of one-year survivors and non-survivors.

Characteristic	Alive at One Year (N = 140)	Dead at One Year (N = 60)	*p*-Value
Age, mean (SD)	82.8 (7.6)	86.2 (7.8)	0.003
Male sex	68 (49,0)	34 (56,7)	0.29
Female sex	72 (51,0)	26 (43,3)	
BMI (kg/m^2^)	26.1 (5.3)	23.3 (5.5)	0.0006
Comorbidities			
Diabetes mellitus	29 (21)	11 (18)	0.76
COPD	23 (16)	12 (20)	0.59
Cancer	11 (8)	6 (10)	0.59
CHF	30 (21)	15 (25)	0.62
CAD	20 (14)	11 (18)	0.47
CVD	108 (77)	48 (80)	0.65
Dementia	26 (19)	21 (35)	0.012
Chronic Renal Failure	31 (22)	16 (27)	0.62
Severity scores			
CURB-65			0.0002
1	32 (23)	9 (15)	
2	66 (47)	18 (30)	
3	38 (27)	23 (38)	
4	4 (3)	10 (17)	
SOFA	2.3 (1.4)	2.9 (2.1)	0.014
Geriatric scores			
FIM (N = 171)	76.3 (32.0)	64.5 (26.0)	0.009
CIRS-G	22.2 (6.5)	25.5 (5.2)	0.0003
MNA (N = 178)	8.9 (2.6)	6.8 (3.0)	<0.0001
Laboratory findings			
Urea (mmol/l)	8.9 (4.8)	10.4 (6.4)	0.033
Albumin (g/l) (N = 193)	35.2 (5.3)	34.2 (5.7)	0.13
PaO2 (kPa)	10.2 (4.3)	10.7 (5.2)	0.25
FiO2	0.27 (0.12)	0.30 (0.14)	0.10
Hemoglobin (g/L)	123 (19.3)	124 (16.9)	0.30
WBC (G/L)	11.7 (4.8)	11.1 (5.9)	0.24
CRP (mg/L)	128 (103)	94 (73)	0.01
NT-proBNP (ng/l) (N = 170)	2537 (2651)	3402 (3032)	0.035

Data presented as n (%) or mean (SD). The number of patients (N) is 200 unless otherwise stated. BMI: body mass index; CAD: coronary artery disease; CIRS-G: cumulative illness rating scale—geriatric; CHF: chronic heart failure; COPD: chronic obstructive pulmonary disease; CRP: C-reactive protein; WBC: white blood cells, CURB-65: confusion urea respiratory rate blood pressure, 65-years old; CVD: cardio-vascular disease; FIM: functional independence measure; FiO2: fraction of inspired oxygen; MMSE: mini mental state examination; MNA: mini nutritional assessment; NT-proBNP: N-Terminal prohormone of brain natriuretic peptide; PaO2: partial pressure of arterial blood oxygen. SOFA: sequential organ failure assessment score.

**Table 2 geriatrics-06-00112-t002:** Association between scores evaluating comorbidities, malnutrition and functionality and 1-year mortality (univariate and multivariate analysis).

**A. Univariate.**			
**Variable**	**OR**	**95% CI**	** *p* **
CIRS-G	1.09	1.06–1.12	0.001
MNA	0.76	0.71–0.81	<0.001
FIM	0.98	0.97–0.99	0.02
**B. Multivariate**			
**Variable**	**OR**	**95% CI**	** *p* **
CIRS-G	1.08	1.01–1.15	0.014
MNA	0.83	0.71–0.96	0.012
FIM	0.99	0.97–1.0	0.269

**Table 3 geriatrics-06-00112-t003:** Sensitivity, specificity, positive and negative likelihood ratio and diagnostic odds ratio (DOR) values for 1-year mortality according to CIRS-G, MNA and FIM at the best cut-off values (computed with Youden index).

Variable	Sensitivity	Specificity	LR+	LR−	DOR(LR+/LR−)
CIRS-G ≥ 26	0.59	0.70	1.94	0.59	3.27
MNA ≤ 8	0.71	0.65	2.0	0.45	4.46
FIM ≤ 63	0.58	0.52	1.21	0.81	1.50

## Data Availability

On demand. Code availability: SECUTRIAL.

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
