# Peer review of "How Do Geriatric Scores Predict 1-Year Mortality in Elderly Patients with Suspected Pneumonia?"

_geriatrics, 2021, doi:10.3390/geriatrics6040112_

Round 1

Reviewer 1 Report

This article deals with an interesting topic in geriatrics, is well written and the methods are adequate.

I have only minor revisions to suggest :

1- Please improve the image quality of second ROC curve in figure 1.

2- Please briefly explain in the introduction how the scores tested in this study were chosen.

3- I suggest to specify in the discussion that the comprehensive geriatric assessment cannot be reduced to these 3 tests.

4- MNA is only one aspect of malnutrition evaluation. Do the authors had access to albumin or weight, BMI, or weight loss ? Such tools are also expected to have high long-term prognostic value in this population

5- The authors conclude : "The Cumulative Illness Rating Scale-Geriatric (CIRS-G) and Mini-Nutritional Assessment (MNA) were found to be promising prognostic tools for assessing long-term mortality in a cohort of elderly people suffering from suspected pneumonia"

I wonder from this article if these tests have any particular predictive value in pneumonia or if such results would have been found in any geriatric cohort (i.e. without pneumonia). This point could be discussed.

Reviewer 2 Report

  1. There are no strong evidence to support the impact of comorbidities, malnutrition, and functionality to predict the mortality in older patients having a suspicion of pneumonia in the section of introduction.
  2. Line 120, what is the CURB-65 severity score, it needs to be described in detail, and states the reasons to be added into the logistic regression model.
  3. The items of laboratory findings in Table 1 are not consistent with the illustration in lines 85-86.
  4. Line 159, Tables 2 , A2 and A3, please add the Appendix.
  5. This is an interesting topic. The writings in the section of results need to be more readable and clear.

Round 2

Reviewer 2 Report

This revised manuscript has been much improved.